# Unidirectional rotation of micromotors on water powered by pH-controlled disassembly of chiral molecular crystals

Itai Carmeli[1,11], Celine M. Bounioux[1,11], Philip Mickel[2,11], Mark B. Richardson[3], Yael Templeman[4], Joel M. P. Scofield[5], Greg G. Qiao [5], Brian Ashley Rosen[1], Yelena Yusupov[1], Louisa Meshi [4], Nicolas H. Voelcker [6], Oswaldo Diéguez[1,7], Touvia Miloh[8], Petr Král [2,9] ✉, Hagai Cohen[10] & Shachar E. Richter[1] ✉

Biological and synthetic molecular motors, fueled by various physical and chemical means, can perform asymmetric linear and rotary motions that are inherently related to their asymmetric shapes. Here, we describe silver-organic micro-complexes of random shapes that exhibit macroscopic unidirectional rotation on water surface through the asymmetric release of cinchonine or cinchonidine chiral molecules from their crystallites asymmetrically adsorbed on the complex surfaces. Computational modeling indicates that the motor rotation is driven by a pH-controlled asymmetric jet-like Coulombic ejection of chiral molecules upon their protonation in water. The motor is capable of towing very large cargo, and its rotation can be accelerated by adding reducing agents to the water.

Biological molecular motors, such as ATP synthase, bacterial flagella, molecular pumps, and others[1–4], can efficiently transform the free energy from their nonequilibrium surroundings into unidirectional translational and rotational motions. Analogous synthetic molecular motors, often formed by asymmetric microparticles[5–9], can be self-propelled by various ratchet-like mechanisms using the energy provided by chemical reactions, electromagnetic and vibrational fields, molecular concentration, temperature, and surface tension gradients[10–17]. Such synthetic motors can be applied in energy harvesting at the microscale, molecular cargo delivery, bacterial isolation, and water purification[10,18,19].

The dance of camphor flakes at the air-water interface[20,21], first reported by Heyde[22], is an excellent example of self-propelled motors[23]. Apparently, this motion is driven by gradients of surface tension caused by the uneven dissolution of camphor on the water–air interface. In naturally formed camphor flakes with arbitrary structures and shapes, the rotation direction can randomly flip according to the varying shapes of the disintegrating particles. In contrast, in the Marangoni effect[24], molecules are released from an asymmetric object floating at the air–fluid interface, which leads to the asymmetric formation of regions with elevated surface tensions, causing a clockwise (dextrogyre) or anticlockwise (levogyre) rotation of the object. However, it is not clear if the symmetry and chemistry of released molecules alone can control the direction of motor rotation. In principle, this idea should be feasible under suitable conditions. For example, it was suggested that melting chiral particles and their assemblies could

[1]Department of Materials Science and Engineering, Faculty of Engineering & University Center for Nano Science and Nanotechnology, Tel-Aviv University, Tel-Aviv 6997801, Israel. [2]Department of Chemistry, University of Illinois at Chicago, Chicago, IL 60607, USA. [3]CSIRO Manufacturing, Bayview Avenue, Clayton, VIC 3168, Australia. [4]Department of Materials Engineering, Ben-Gurion University of the Negev, Beer-Sheva 84105 POB 653, Israel. [5]Department of Chemical Engineering, University of Melbourne, Parkville, VIC 3010, Australia. [6]Drug Delivery, Disposition, and Dynamics, Monash Institute of Pharmaceutical Sciences, Monash University, 381 Royal Parade, Parkville, VIC 3052, Australia. [7]The Raymond and Beverly Sackler Center for Computational Molecular and Materials Science, Tel Aviv University, Tel Aviv 6997801, Israel. [8]School of Mechanical Engineering, Faculty of Engineering, Tel Aviv University, Tel Aviv 6997801, Israel. [9]Department of Physics, Pharmaceutical Sciences, and Chemical Engineering, University of Illinois at Chicago, Chicago, IL 60607, USA. [10]Department of Chemical Research Support, Weizmann Institute of Science, Rehovot 76100, Israel. [11]These authors contributed equally: Itai Carmeli, Celine M. Bounioux, Philip Mickel. ✉e-mail: pkral@uic.edu; srichter@tauex.tau.ac.il

perform unidirectional rotations[25]. This can be explained by the de Gennes' theory[26], predicting that freely floating chiral crystals in nonequilibrium conditions could spin unidirectionally.

Here, we demonstrate the unidirectional rotation of chiral motors ("chimots") formed by millimeter-sized silver-organic complexes with random structures and surfaces consisting of pseudo-enantiomeric cinchona alkaloid natural products, (+)-cinchonine ("cin+") or (−)-cinchonidine ("cin−") molecular crystallites (Fig. 1). These surprising observations reveal that the rotation of chimots on a water surface depends on the chirality of the molecules adsorbed on them. This asymmetric rotation is likely driven by the asymmetric release of these chiral molecules (Fig. 1).

Intuitively, we can draw a parallel between the observed phenomena and the Marangoni effect[24], which attributes a unidirectional rotation of an asymmetric motor to its asymmetrical release of molecules. However, this effect does not explain the correlation between the alkaloid chirality and the chimots' rotation direction. Given the macroscopically irregular surfaces of chimots without any obvious unidirectional asymmetry, one may expect the released molecules to generate a random particle motion. However, the presence of chiral molecular crystals on chimot surfaces might introduce a microscopic asymmetry at the surfaces that can affect the release of these chiral molecules.

## Results and discussion

The prepared chimots (see synthesis in "Methods") were thoroughly dried under a nitrogen atmosphere for several days. When carefully placed on the surface of deionized water (pH = 7), the chimots floated since they were supported by surface tension like water spiders. Most chimots immediately started to rotate clockwise or anticlockwise (Fig. 1 and Supplementary Movies 1–4). The direction of rotation for randomly chosen particles exhibited batch-to-batch consistency of 70–80% (see Supplementary Information, Supplementary Tables 1 and 2). Typically, linear velocities of $v = 0.6–3$ mm/s were obtained along circular trajectories with radii of $r = 1–10$ mm. The self-propulsion of chimots usually lasted from several minutes to four hours. This

behavior was reproducible, provided that the chimots were first dried and carefully put at the water–air interface.

Laser confocal optical and scanning electron microscopy (SEM) images (Fig. 2a–d) reveal that chimot have arbitrary macroscopic shapes and somewhat different colors. SEM micrographs indicate that chimots are composed of many microplatelets, covered with a scattered population of molecular nanocrystals and silver-complex nanoparticles (Fig. 2b, d and Supplementary Fig. 4). Powder X-ray diffraction (XRD, Fig. 2e) of the chimots provides very sharp signals associated with alkaloid crystallites at least 200 nm in size. Reference samples of pure organic powder disclose that each molecular crystal has only one crystallographic phase, $P_{21}$ in cin+ and $P_{212121}$ in cin−. In turn, the coexistence of two (or more) phases is observed in the chimot complexes: (i) domains identical to the pure organic structure and (ii) a phase assigned to the complexation of the chimots with silver, which shows no detectable traces of a pure silver phase.

Similarly, X-ray photoelectron spectroscopy (XPS, see Supplementary Discussion) indicates the existence of non-metallic silver, together with the typical fingerprints of cin+ or cin− molecules. A large excess of organic molecules is quantitatively indicated by the N/Ag XPS atomic ratios, which corroborates the XRD. The release of molecules from the spinning particles and the formation of Ag-free monolayers on the water surface was further verified by XPS (see Supplementary Discussion) and independently substantiated by time-dependent optical absorption and surface tension measurements (Fig. 2f, g and Supplementary Fig. 5, respectively).

Importantly, our pH-dependent study indicates that chimots rotate at natural (pH = 7) and acidic conditions (pH < 7), but they do not rotate under highly basic conditions (pH = 10, see Supplementary Figs. 6 and 7). However, control experiments performed in ethanol, methanol, and isopropanol did not unveil any consistent motion of chimots. Only slow rotation was found in ethylene glycol, around 0.2 cycles/min. Electrophoretic interrogation at pH = 7 revealed that chimots are negatively charged. This charge can partly originate from the passivating cysteine molecules residually present on the chimot

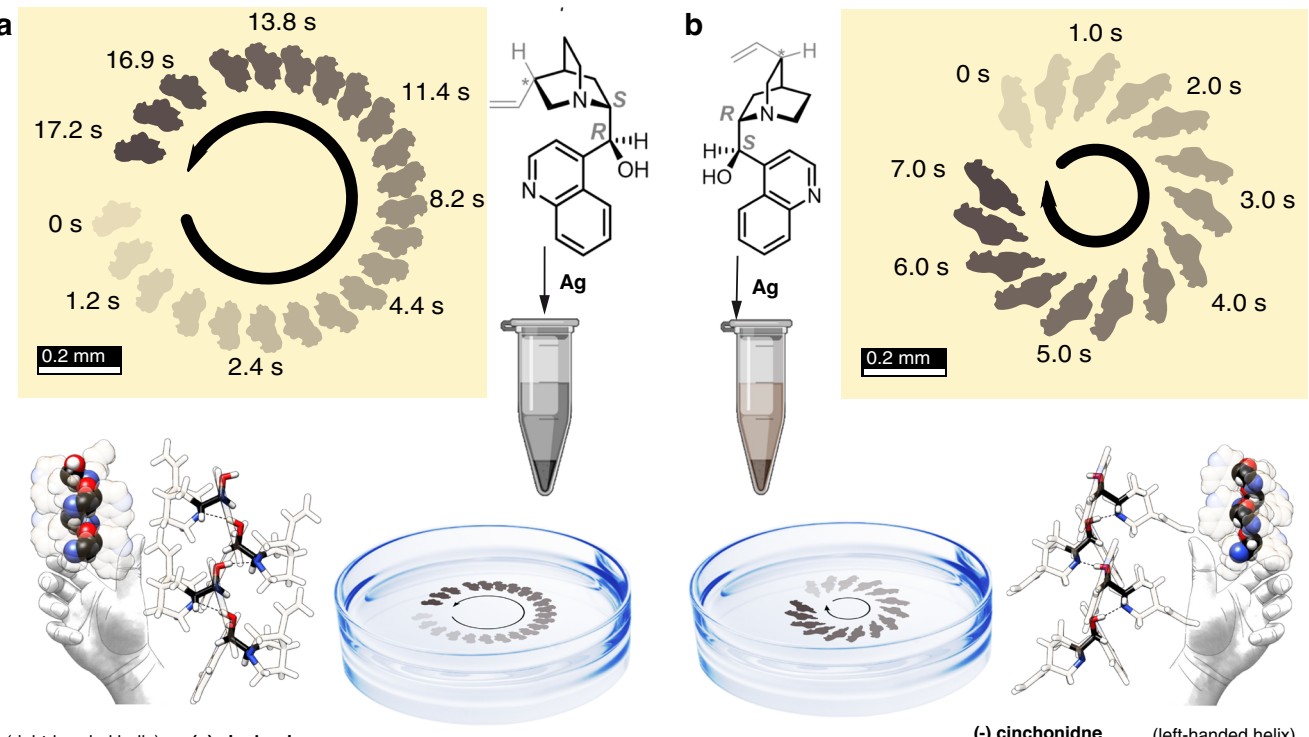

**Fig. 1 | Chimot synthesis and operation.** The synthesis process. Cin + (**a**) and cin− (**b**) are complexed with silver and placed on the water surface. Representative time-lapsed optical images of the chimots are presented.

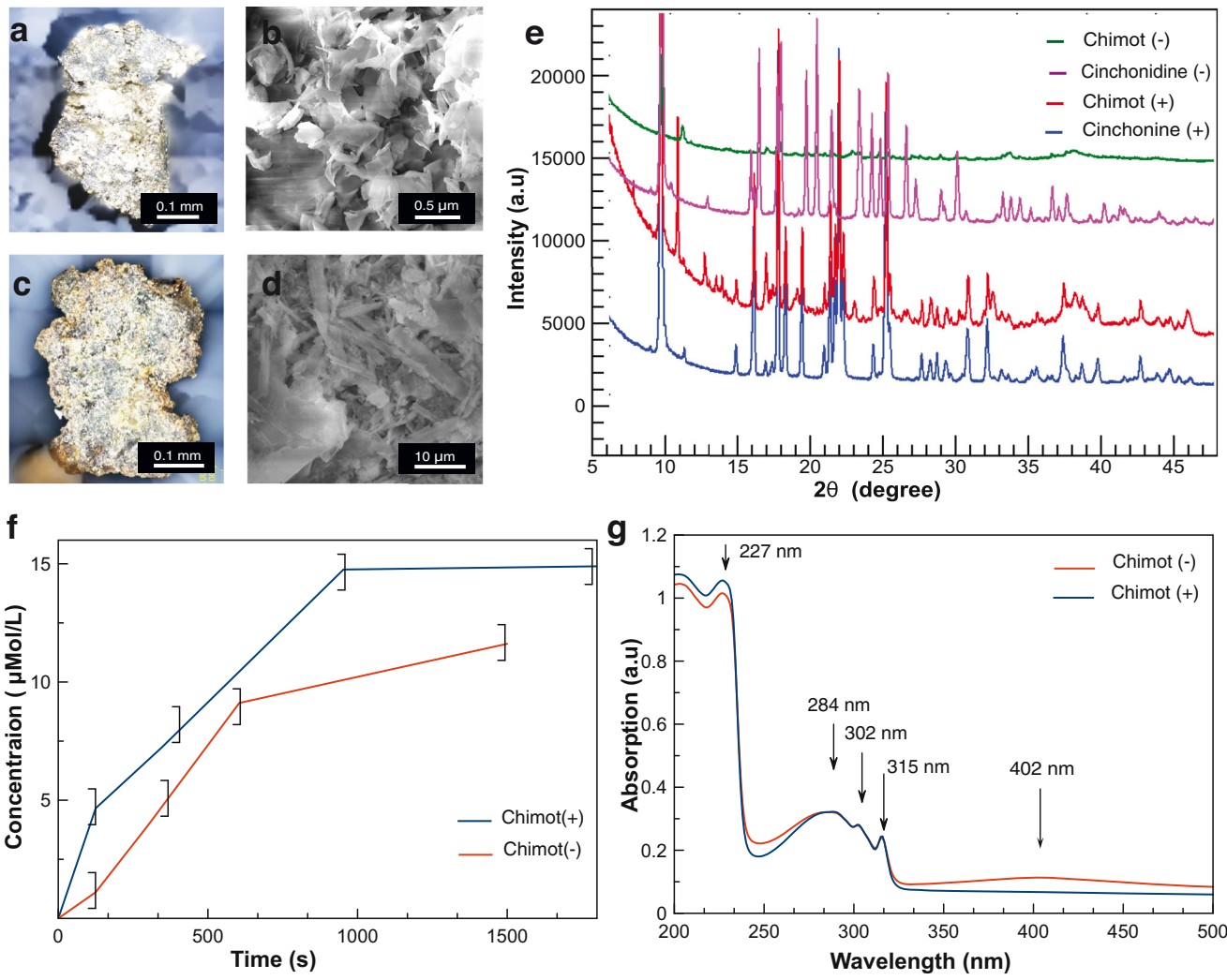

**Fig. 2 | Microscopy and spectroscopy characterization of chimots. a–d** Laser confocal microscopy and scanning electron microscopy images of representative (**a**, **b**) chimot+ and (**c**, **d**) chimot−. **e** X-ray diffraction patterns of reference pure organic samples (blue and pink) and corresponding Ag–cin complexes (red, green) suggest the coexistence of at least two separate phases in the chimot particles.

**f** Time-dependent concentration changes of cinchona alkaloids, determined by UV–Vis spectroscopy of the water–air interface, during swimming of the chimots (evaluated for λ = 315 nm). **g** UV–Vis spectra of the water, taken after the rotation, show full consistency with literature spectra of these alkaloids (main absorbance bands are indicated).

surfaces. However, depending on pH, the solvated chimot surfaces can be additionally charged by protonation of the alkaloids.

To provide clues about the mechanisms causing these chimot rotations, we noticed that chimots often underwent a measurable mass loss, with small particles sometimes seen breaking away from their surfaces. Nevertheless, gas chromatography measurements did not indicate gas generation during the process. Although it became clear that the alkaloid molecules released at the water–air interface most likely fuel the rotation of chimots, it was not clear how their chirality controls the rotation direction. Consisting of both hydrophilic and hydrophobic groups, these molecules preferentially float on the water; hence when the surface is fully covered, a further release is practically terminated, and the propelling motion of chimots should stop. To test this fact, we let particles cease their motion after sufficient time and demonstrated restoration of their rotation up to 10 times upon transfer to freshwater. The activated release of molecules is also supported by temperature-dependent measurements indicating a minimum temperature for the chimot rotation at ~8 °C (see Supplementary Figs. 8 and 9).

In order to further reveal the nature of the observed phenomena, we performed theoretical modeling and atomistic molecular dynamics simulations[27–29] to investigate the stability of chiral molecules within

different crystal facets. We modeled cin molecular crystals with six different facets submerged in water (Fig. 3a). Both cinchona alkaloids have basic nitrogen sites with pKa = 8.43 and 4.25 of the quinuclidine and aromatic nitrogens, respectively. At pH = 10, the molecules are neutral; at pH = 7, the quinuclidine nitrogen is protonated (positively charged); at pH = 4, both nitrogens are protonated (positively charged). Following these conditions, we simulated the molecular crystal at (1) pH = 10, where the entire crystal is neutral, and (2) pH = 7, where the exposed quinuclidine nitrogens of the outermost layer of the crystal are protonated (positively charged). On each facet, only exposed nitrogens can be protonated. Therefore, some facets are completely uncharged while others are only partially charged, as schematically visualized in Fig. 4a by the red-colored charged molecules.

To reveal how the cin molecules can be released from their crystals, we examined the interaction energies of facet-embedded cin molecules with the rest of the crystal and with water in which the crystal is submerged (fixed). Figure 3b shows the calculated interaction energies between molecules embedded in different facets (charged and uncharged) and the rest of the (fixed) crystal (top) and water (bottom). At pH = 10, we found that all coupling energies with the crystal are negative but smaller than those with water, indicating that

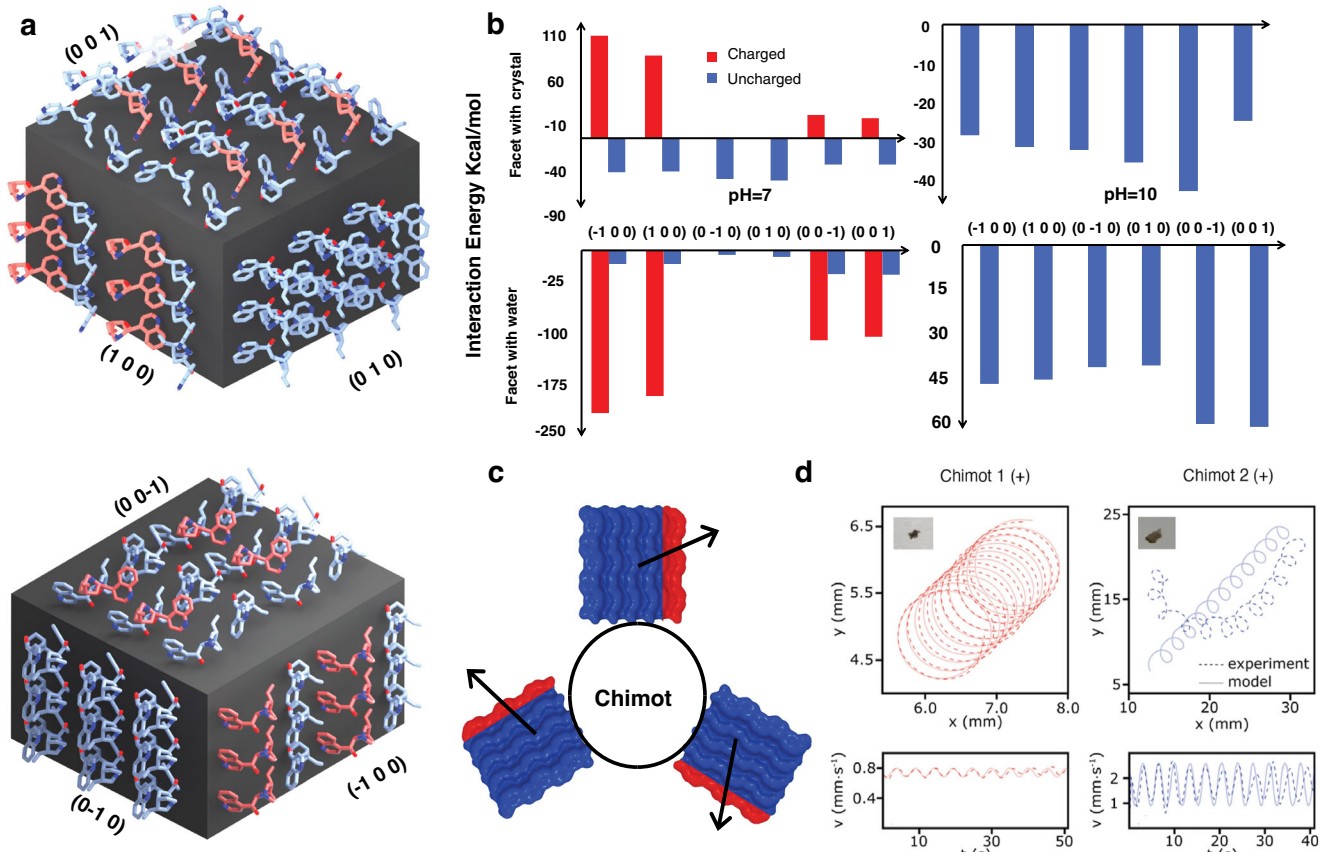

**Fig. 3 | Model and simulations. a** Visualization of cinchonidine crystals with each facet labeled by Miller indices. Red molecules indicate a positive charge at pH = 7. Blue molecules indicate a neutral charge at pH = 7. **b** Interaction energy of facet-embedded cin molecules with water (bottom) and the bulk cin crystal (top), calculated with the NAMDEnergy plugin. At pH = 7 (left) and 10 (right), uncharged molecules exhibit weak attractive interaction energies with water, and stronger attractive interaction energies with the crystal, indicating their retention in the facet. At pH = 7, charged molecules exhibit strong, attractive interaction energies

with water and strong repulsive interaction energies with the crystal, indicating their ejection from the facet. Under both conditions, there is a preference for which facets are most likely to maximize exposure to water. In addition, there is an asymmetry between opposing facets' release of molecules from the surface.
**c** Schematic showing the adsorption of cin crystals to the chimot and the rotation-driving dynamics emerging from the chiral symmetry of the crystallite.
**d** Comparison between experimental (dashed lines) and a Stokesian hydrodynamic rotation model (solid lines) for two different particles.

the molecular crystal might slowly dissolve. In contrast, at pH = 7, the uncharged molecules still exhibit negative binding energies with both the crystal and the water, but these energies are much smaller, showing faster disintegration. However, the charged molecules have large positive repulsion energies with the crystal (charged molecules around) and large attractive energies with water. Therefore, upon their charging at low pH, these molecules should be Coulombically ejected from the crystal and provide the observed rotation-driving force.

Notice that all six facets of the chiral cin crystals are different. One of the facets should have the largest likelihood of binding to the chimot core, while other facets are likely submerged in water. Although these crystallites can be randomly oriented on the chimot surface, the most relevant crystallites are positioned around the slightly submerged circumference of chimots, where they asymmetrically release molecules from their water-submerged facets (possibly (1 0 0) and (−1 0 0)). This asymmetrical release of charged alkaloids apparently breaks the rotation symmetry of chimot driving. Figure 3c schematically shows how molecules, asymmetrically released from individual crystals, transfer angular momentum to chimots, which results in a unidirectional rotation. The actual forces acting on each microcrystal are caused by a concentration difference (micro-gradient) of cin molecules dissolved from different facets (see Supplementary Information). Therefore, the opposite cin enantiomer should rotate chimots in the opposite direction, as observed experimentally. Upon dissolution of

the cin crystals, the film on the surface should contain little to no silver, in agreement with our XPS analysis. Note also that the MD simulations depicted in Fig. 3 indicate that the releases molecules tend to accumulate at higher concentration on one side of the microcrystal, resulting in a diffusiophoretic driving exerted on the floating chimots due to concentration gradients.

To better understand the observed spinning of chimots, we simulated their motion using a simplified Stokesian hydrodynamic model (see Supplementary Information). The viscous forces and torques acting on the particle were estimated by comparing the experimental and model dynamics results (see Supplementary Information for details). A similar approach has been employed to calculate (Fig. 3d) the circular trajectories of chiral micron-sized particles[5,25]. The hydrodynamical model qualitatively agrees with the observed chimot rotation phenomena. We have also used the above MD simulation results to estimate the forces acting on chimots (see comment in the Supplementary Information). Given the lack of statistics of microcrystal distribution, the obtained forces are several orders of magnitude larger than those used in hydrodynamic model that fits the experimental observations. This shows that the driving could be significantly enhanced if the system parameters are further optimized.

To increase the release rate of cin molecules, we have introduced an active chemical-reducing agent, $NaBH_4$, in water. We hypothesize that this molecule reduces the silver cation, thus accelerating the

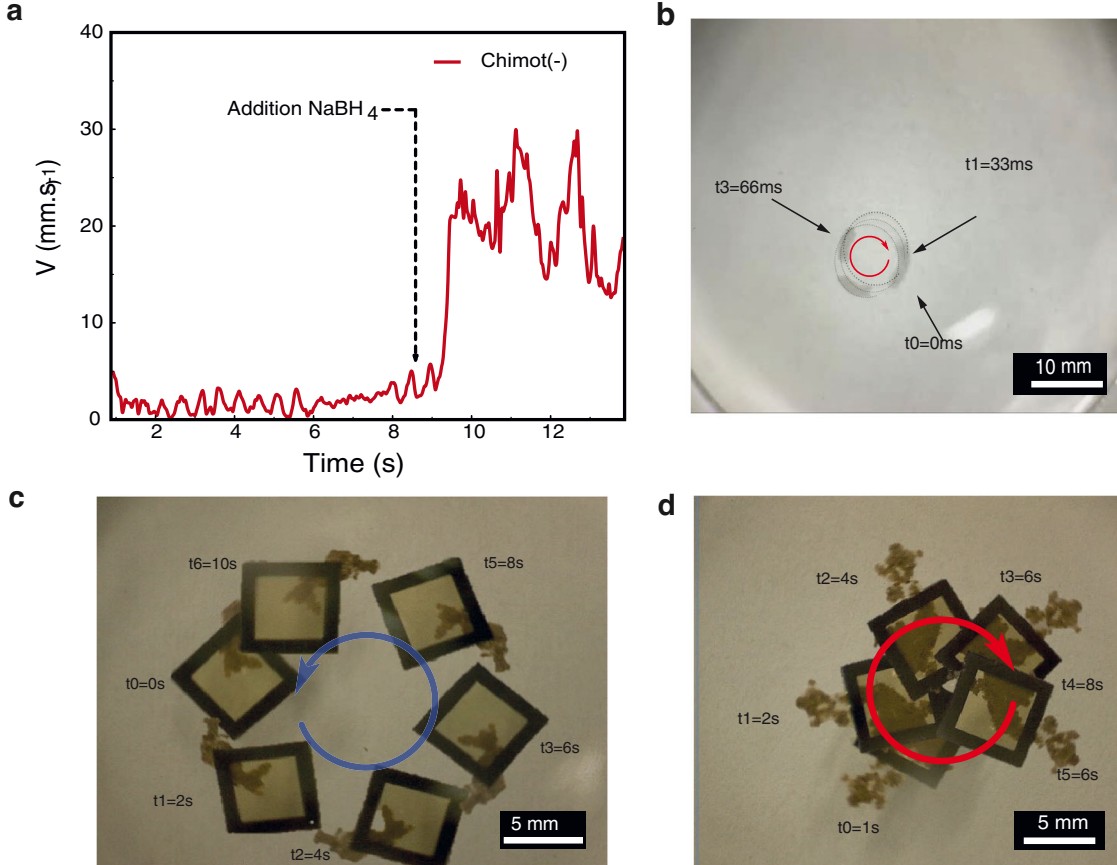

**Fig. 4 | Dynamics of chimots. a** Acceleration of the chimots and towing characteristics, including recorded values of chimot+ acceleration after adding NaBH₄. An order of magnitude increase in the motor's rotational velocity is observed. **b** A timelapse image of representative chimot motion after adding the reduction agent. **c**, **d** Timelapse images showing the chiral motion of (**d**) cin+ and **e** cin− chimots carrying an X40 cargo.

release of cin molecules and individual crystals to the water surface. Figure 3a, c reveals that the chimot rotation accelerates in the presence of such a reducing agent (see Supplementary Movies 7 and 8 and Supplementary Discussion for supporting XPS analysis), featuring translational velocities up to $v = 30$ mm/s. Therefore, the current cin-based particle is one of the fastest known self-propelling organic micromotors (Fig. 4b, see also Supplementary Movies 6 and 7 and Supplementary Information).

One of the prominent properties of micro-swimmers is their remarkable capability as cargo carriers[30]. This is a key issue in biomedical transport and drug delivery applications, as micro-swimmers can only sustain a limited load before losing their propelling speed. Figure 4d, e shows the result of a typical towing experiment with the present chimots. A silicon frame (SF) with a weight ratio of SF/chimot ~40 is placed on top of "+" and "−" chimots. As evident from Fig. 4 and the recorded velocities, the chimots' motion was found to be rather persistent (i.e., lasting for up to 4 h), exhibiting relatively high speeds, up to few tenths of mm/sec; much faster than any of the previously reported cases (see also Supplementary Movies 5 and 6 and relevant refs. in Supplementary Information).

In summary, we demonstrated that metal-organic complexes of random shapes and adsorbed crystallites of chiral molecules can act like molecular motors providing unidirectional rotation on the water surface. Our experimental and computational results provide solid evidence that this pH-controlled rotation is Coulombically driven by an asymmetrical jet-like release of charged chiral molecules and crystallites at the water–air interface. The direction of chimot motor rotation is solely determined by the handedness of adsorbed chiral molecules, while its rotation speed is controlled by the release rates of chiral molecules, which is dependent on pH and the presence of reducing agents. We observe that chimots can carry relatively large cargoes, revealing a large potential of this intricate physical phenomenon for future exciting applications.

## Methods
### Materials
Cinchonine >98%, cinchonidine 96%, sodium hypophosphite hydrate, sodium borohydride, ethylene glycol, were purchased from Sigma Aldrich. Silver nitrate 99.9% was obtained from STREM chemicals. Trisodium citrate and sodium decosulfate (SDS) were purchased from Merck.

### Methods
**Synthesis**. Stock solutions of the reaction compounds were prepared: 18 mg of trisodium citrate in 12 ml deionized water (DI), 34 mg of sodium hypophosphate in 12 ml DI, 20 mg of sodium dodecyl sulfate (SDS) in 2 ml DI, and 36 mg of silver nitrate in 12 ml DI. Next, 6 mg of cinchonine/ cinchonidine were partially dissolved in 2 ml DI water containing 100 μL of SDS and stirred for 10 min, then added 7 ml of DI to 1 ml of DI silver nitrate stock solution. Then, the mixture was stirred for two hours in the dark. In the next step, 1 ml of citrate and 1 ml of sodium phosphate stock solution were added drop by drop to the mixture. The reaction vessel was left to stir for eight consecutive days in the dark until the appearance of chimot aggregates. Then, the solution was centrifuged and washed a couple of times with DI water to recuperate the materials, which were dried for several days under a nitrogen atmosphere. The dry powder was stored in Eppendorf with nitrogen gas for further use.

**Rotation experiments.** A PET glass or Teflon Petri dish (3, 5, and 7 cm in diameter) was filled with DI water. A single particle was carefully positioned at the air/water interface, and its motion was recorded with a CCD camera mounted on a microscope or by iPhone 6plus camera. Analysis of the velocity was done by an image-processing protocol using Matlab software.

## Characterization

**UV–Vis.** UV–Vis measurements were carried out with a CARY 5000 UV–Vis–NIR spectrometer.

**Optical characterization.** Optical images were taken using an Olympus BX51 microscope with and without a polarization filter. SEM images were done using FESEM or JSM-6300 instruments. TEM investigation was carried out using a 200 kV JEOL JEM 2011, FasTEM, and a JEOL JEM-2100F TEM operating at 200 kV equipped with JED-2300T EDX.

**XPS.** XPS measurements were performed on a Kratos AXIS-Ultra DLD spectrometer, using a monochromatic Al kα source at a power ranging between 15 and 75 W and detection pass energies of 20–80 eV. The pressure in the analysis chamber was kept below $10^{-9}$ torr. Ar-ion sputtering was used only at fine levels to follow the early stages of adsorbed molecules' removal from the highly corrugated surface morphology. Marked differential charging was typically encountered at the platelet aggregates, which normally complicates the interpretation of XPS results. Here, however, differential charging was exploited to differentiate between signals originated from different domains, which enhances the capabilities of the (large area) XPS probe significantly[31,32]. Samples were prepared for the XPS analyses by carefully lifting Si wafer substrates that were pre-positioned at the bottom of the reaction cell. Thus, platelets floating on the liquid surface could be captured at various stages of the spinning process. In order to selectively measure those films of molecules released from the platelets to the water, we removed all (observable) aggregates from the liquid surface prior to lifting the Si wafer. Macroscopic fragments of the Langmuir film could thus be XPS-analyzed[31,32].

**Surface pressure.** The surface pressure was measured by the Wilhelmy plate method, using a sensor from Nima Technologies. Due to the small difference of surface pressure, the experiment was done in a Petri dish (3.5 cm diameter) and measurements took place at a rate of two samples per second. The water's surface was first cleaned using a vacuum aspirator to remove dust and other possible materials from the surface. Samples of chimots or camphor particles were used for each experiment and placed on the surface simultaneously as the surface measurements were started. XRD scans spectra were taken by means of a Bruker D8 DISCOVER diffractometer using CuKα radiation.

**Hydrogen content measurements.** In total, 30 mg of chimots were added to 20 mL of water in a sealed vial. The headspace of the vial was sampled by an SRI gas chromatograph using an automatic sampling valve 1 mL in volume. Hydrogen was monitored using a nitrogen carrier gas, a MS 5 A column and a TCD detector. Hydrocarbons and CO were monitored with MS 5 A and Hayesep D columns and an FID detector with a methanizer at its inlet. Throughout the experiment, no gas was generated above the 10-ppm detection limit (1-ppm for CO).

**Molecular dynamics simulations.** The systems were simulated using NAMD 2.13[25,28] and the CHARMM 36[26,27,29] protein force field, with the water bath in the NVT ensemble at a temperature of $T = 310$ K. The MD simulations were conducted with Langevin dynamics ($\gamma_{lang} = 1$ ps$^{-1}$) in the NPT ensemble at a temperature of $T = 310$ K and a pressure of $p = 1$ bar. The particle-mesh Ewald (PME) method was used to evaluate Coulomb coupling, with periodic boundary conditions applied. The time step was set to 2 fs. The long-range van der Waals and Coulombic coupling were evaluated every 1 and 2 timesteps, respectively. A 5- by-5-by-5 crystal of cin− was minimized for 200,000 steps and warmed for 2000 steps[33,34]. The crystal was equilibrated for 1 ns with all non-hydrogen cin- atoms constrained (1 kcal (mol Å)$^{-1}$). The innermost cin− molecule on each facet was then unconstrained, and the crystal was equilibrated for 5 ns.

**MMGB-SA calculations.** The molecular mechanics generalized Born-surface area (MMGB-SA) method[35,36], was used to estimate the free binding energies between embedded facet molecules with their crystals, as well as entire crystal facets to nearby solvent molecules. The free energies were estimated from separate MMGB-SA calculations for three systems (e.g., facet, solvent, and facet + solvent) in configurations extracted from the MD trajectories of the whole complex in the explicit solvent. The MMGB-SA free energies of the extracted configurations of the systems were calculated as

$$G_{tot} = E_{MM} + G_{solv-p} + G_{solv-np} - T\Delta S_{conf}, \tag{1}$$

where $E_{MM}$, $G_{solv-p}$, $G_{solv-np}$, and $T\Delta S_{conf}$ are the sum of bonded and Lennard-Jones energy terms, the polar contribution to the solvation energy, the nonpolar contribution, and the conformational entropy, respectively. The $E_{MM}$, $G_{solv-p}$, and $G_{solv-np}$ terms were calculated using the NAMD 2.13[27,29] package generalized Born implicit solvent model[37], with the dielectric constant of water $\varepsilon = 78.5$. The $G_{solv-np}$ term for each system configuration was calculated in NAMD as a linear function of the solvent-accessible surface area (SASA), determined using a probe radius of 1.4 Å, as $G_{solv-np} = \gamma$ SASA, where $\gamma = 0.00542$ kcal mol$^{-1}$ Å$^{-2}$ is the surface tension. The $\Delta S_{conf}$ term was neglected since the entropic contribution nearly cancels when considering binding of such similar structures. Since the $G_{tot}$ values are obtained for configurations extracted from the trajectories of complexes, $G_{tot}$ does not include the free energies of reorganization; the correct free energies of binding should consider configurations of separately relaxed systems. The approximate binding free energies of the studied complexes were calculated as:

$$\langle \Delta G_{MMGB-SA} \rangle = \langle G_{tot}(crystal - facet\ molecule) - G_{tot}(crystal) - G_{tot}(facet\ molecule) \rangle, \tag{2}$$

where the ⟨averaging⟩ is performed over 5 ns trajectories.

## Data availability
The authors declare that all data supporting the findings of this study are available within the paper and its supplementary information files.

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

## Acknowledgements

We thank Professor Meir Lahav for the fruitful discussions. NV and SR thank the Monash-TAU grant for financial support.

## Author contributions

I.C. discovered the effect. I.C., C.M.B., and Y.Y. synthesized the materials and performed the rotation measurements. Y.T. and L.M. performed the electron diffraction measurements and analysis. H.C. performed the XPS measurements and analysis. J.M.P.S. performed the surface tension experiments. G.G.Q. mentored the surface tension experiments. B.A.R. performed the hydrogen generation measurements. P.K., P.M., O.D., and T.M. performed the molecular dynamics and model calculations. S.R., M.B.R., H.C., T.M., O.D., P.M., C.P., C.M.B., and N.H.V. constructed the mechanism. All authors wrote the paper.

## Competing interests

The authors declare no competing interests.
