## [Peer Review File · Nature Communications]

Unidirectional Rotation of Micromotors on Water Powered by pH-Controlled Disassembly of Chiral Molecular CrystalsReviewers' comments:

Reviewer #1 (Remarks to the Author):

In this paper a self-propelling motor at the water-air interface driven by the release of molecules is studied which exhibits chiral self-motion. I do not recommend this paper for publication in Nature Communications due to the following reasons:

- 1) The authors claim that this is a new type of self-propelling motor and that "artificial analogs are less abundant". But there are many objects on an interface which are self-propelled (some examples which are not cited and maybe overlooked by the authors in this vast literature field are: Scholz et al, Nature Comm. 9,931 (2018); Frenkel et al, Scientific Reports 7, 3930 (2017); C Krüger et al, Physical review letters 117, 048003 (2016)). Just proposing a macroscopic system which self-circles at an interface is not enough. The rotation sense is not tunable during the motion externally (as could be done with circularly polarized light, for example) but it is inherent in the preparation of the particles. What is really the new message here remains obscure. Even the aspect of towing a cargo has been explored in much more quantitative way before. Hence there is no real novel aspect in the paper (except from the quite obvious finding that the chirality of the motor depends on the chirality of the molecules used).
- 2) The paper is inconsistent. The model used is in the overdamped Stokes regime which is the regime of small Reynold numbers, see equation 1] in the SM (by the way Equation [2] is wrong and has probably a missing ")"). But on the other hand, the particles are macroscopic (millimetric size). So the Navier-Stokes equations should be considered not the overdamped linearized Stokes equations. Hence the theory needs a significant upgrade.
- 3) There are statements that molecules are "ejected" and the picture is that "tangential momenta are exchanged". How can this happen in a liquid (water) without taking solvent flow effects into account?
- 4) Maybe the self-propulsion mechanism has a significant phoretic part. Why can phoretic driving mechanisms be excluded?
- 5) In Figure 3C and 3D 2d trajectories are shown but not analyzed. Are these real mathematical helices?

Reviewer #2 (Remarks to the Author):

The authors present a system where molecular chirality and relative chemical reactions result in a large-scale chiral active motion. The active particles are made of silver complex with chiral molecules, which placed at the air-water interface are gradually released producing a linear and angular momentum on the particle, which moves of chiral active motion. The same particles are used also to carry around some cargos. The idea of the work is interesting especially when it draws a connection between nanoscale molecular chirality and large-scale chiral active motion. However, there are several points that are unclear and need to be addressed:

1) It's not clear how the chirality of the silver particle itself influence this overall process. It'd be necessary to study this aspect theoretically extending the theory as well as experimentally with increased statistics and with silver complexes with controllable shapes.

2) It's not clear how this work can lead to the concrete applications, which are mentioned in the article. It'd be necessary to either demonstrate one such application, or to indicate a clear pathway towards such application.

These issues would need to be addressed very carefully with new theory and experiments before I can feel confident to recommend publication in Nature Communications.

Reviewer #3 (Remarks to the Author):

The paper by Carmeli et al. reports on self-propulsion of millimeter-sized objects at the water-air interface. The motion of such artificial motors, which are made of silver-organic complexes, results from the release of alkaloid molecules into the interface and completely stops when the concentration of alkaloid molecules saturates on the water surface. Depending on the type of the ejected molecules (cinchonine or cinchonidine) the motors can perform a persistent circular motion with a clockwise or anti-clockwise sense of rotation. Remarkably, the typical velocities achieved by the motors are of the order of several millimeters per second, whereas they are able to tow heavy cargo particles. Although the idea of designing efficient self-propelling motors have been receiving an increasing attention in various fields of both fundamental and applied research, the analysis and the results presented in this manuscript are scarce and do not provide a clear explanation of how the shape and size of the motors or the chirality of the released organic molecules can be employed to perform specific tasks at fluid-fluid interfaces. In addition, I have several concerns with regard to the data analysis, the modelling, and the interpretation of the results, which are listed below:

1) The paper lacks solid evidence of the physical origin of the rotational motion of the motors. The authors attribute the sense of rotation of the motors to the chirality of the ejected alkaloid molecules. However, a close inspection of table S1 of the Supplemental Information reveals that a significant fraction of the motors of each batch does not move as expected. In addition, the number of tested motors per batch (at most 15) is too small to correlate the rotational motion to the chirality of the molecules. The authors should at least improve the statistics of their experiments in order to draw reliable conclusions.

2) What is the role of temperature in the circular motion of the motors? For camphor flakes self-driven by Marangoni flows, it has been reported that the temperature strongly affects the frequency of revolution of the motion along the circular orbits [Nakata, S. et al., *Langmuir* 13, 4454–4458 (1997)]. The authors must specify under which temperature conditions they carried out their experiments.

3) Did the authors investigate the effect of the motor shape on the resulting circular motion, e.g. on the radius and velocity?

Without this kind of analysis, it is impossible to disentangle the role of the Marangoni flow, the chirality of the motor due to its own shape and the chirality of the released molecules on the torques that give rise to persistent rotational movement.

4) The details about the model are not clear. In the SI, the authors mention briefly that they used a generalized Stokes approach based on the grand-resistance matrix to numerically find trajectories that resemble the experimental ones. However, they do not explain how to obtain the corresponding

matrix elements, and how they are related to the alleged microscopic mechanism that produces the self-propulsion and self-rotation of the motors. In that sense, the model does not provide any useful information that elucidates how to exploit the proposed mechanism for particle steering, cargo towing, etc.

5) The authors must thoroughly check the writing of the paper. They often refer to the SI for many details about the experiments and the model, but the needed information is missing there or is not clearly explained. Some references of the bibliography do not correspond to the main text (e.g., 21, 22 and 23). The SI contains a large amount of figures (S1, S2, S3, S5) that are never mentioned in the paper, and some of the figure and equation numbers are wrong (e.g. Figure X, Eq. A11, etc.)

In view of the previous points, I cannot recommend this paper for publication in Nature Communications in its current form, unless the authors make a major revision taking into account the remarks mentioned above.

Although we cannot offer to publish your paper in Nature Communications, the work may be appropriate for another journal in the Nature Research portfolio. If you wish to explore suitable journals and transfer your manuscript to a journal of your choice, please use our <https://mts-ncomms.nature.com/cgi-bin/main.plex?el=A7S3BtpN1A7KMLh6X5A9ftdN7KboConWMx98CgVc2aGAgZ> manuscript transfer portal. If you transfer to Nature-branded journals or to the Communications journals, you will not have to re-supply manuscript metadata and files. This link can only be used once and remains active until used.

All Nature Research journals are editorially independent, and the decision to consider your manuscript will be taken by their own editorial staff. For more information, please see our http://www.nature.com/authors/author_resources/transfer_manuscripts.html?WT.mc_id=EMI_NPG_1511_AUTHORTRANSF&WT.ec_id=AUTHOR manuscript transfer FAQ page. Note that any decision to opt in to In Review at the original journal is not sent to the receiving journal on transfer. You can opt in to *[In Review](https://www.nature.com/nature-research/for-authors/in-review)* at receiving journals that support this service by choosing to modify your manuscript on transfer. In Review is available for primary research manuscript types only.

Prof. Shachar Richter
Department of Materials Science and Engineering
Faculty of Engineering
&
University Center for Nanoscience and Nanotechnology
Tel Aviv University, Tel-Aviv, 69978, Israel

פרופ. שחר ריכטר
המחלקה למדע והנדסה של חומרים
הפקולטה להנדסה
&
המרכז לננומדע וננוטכנולוגיה
אוניברסיטת תל אביב, תל אביב 689978 ישראל

טל: (972)-36405711
פקס: (972)-36405612
srichter@tauex.tau.ac.il

Summary of new features and summary of the additional points of novelty in this manuscript:

1. The effect of pH on the rotation of chimots was experimentally explored and added.
2. This observation allowed us to understand chimot driving by means of molecular dynamics simulations. This new section of the theoretical part is now included in the revised version (Fig. 3). It provides an explanation of the propelling mechanism as a Coulombic ejection of charged chiral molecules causing asymmetrical driving of the chimot rotation.
3. The effect of temperature on the rotation was also investigated.
4. Additional statistical data on rotation was added.
5. The hydrodynamics model was expanded and improved.

For this task experts in molecular dynamics and hydrodynamic contributed new parts to this version

Reviewers' comments:

Reviewer #1 (Remarks to the Author):

In this paper a self-propelling motor at the water-air interface driven by the release of molecules is studied, which exhibits chiral self-motion.

1) The authors claim that this and that "artificial are less abundant". But there are many objects on an interface which are self-propelled (some examples which are not cited and maybe overlooked by the authors in this vast literature field are: Scholz et al, Nature Comm. 9,931 (2018); Frenkel et al, Scientific Reports 7, 3930 (2017); Physical Review Letters 117, 048003 (2016)). Just proposing a macroscopic system that self-circles at an interface is not enough. System which self-circles at an interface is not enough.

Reply: The Referee is correct when it comes to the earlier reports on self-propelled objects. However, we introduce a *different* mechanism, i.e., driven by molecular asymmetry and not by macro-shape characteristics. As such, the present experiments (and theory; see further

Prof. Shachar Richter
Department of Materials Science and Engineering
Faculty of Engineering
&
University Center for Nanoscience and Nanotechnology
Tel Aviv University, Tel-Aviv, 69978, Israel

פרופ. שחר ריכטר
המחלקה למדע והנדסה של חומרים
הפקולטה להנדסה
&
המרכז לננומדע וננוטכנולוגיה
אוניברסיטת תל אביב, תל אביב 689978 ישראל

טל: (972)-36405711
פקס: (972)-36405612
srichter@tauex.tau.ac.il

answers below) are far from 'just an additional example of...'. The translation of molecular properties into this type of **macro**-movement is novel and, in view of its remarkable performance, very appealing as well. It certainly can motivate new fundamental studies and new types of applications.

To the best of our knowledge, this is the first experimental example showing a unidirectional rotation driven by such a mechanism. Significant progress towards its understanding was achieved in our *molecular dynamics* calculations, now included in our revised version. Notably, the papers mentioned by the referee deal with different aspects of molecular motors, such as inertial issues in motors (Nat. Com. 2018), relations to chemical gradients (Sci. Rep. 2018), and theoretical issues of symmetry breaking in motion (PRL). We added all the references suggested by the referee to the current version of our manuscript and provided a broad literature survey.

2) *The rotation sense is not tunable during the motion externally (as could be done with circularly polarized light, foir example) but it is inherent in the preparation of the particles.*

Reply: We would like to thank the referee for this comment. However, this is only partly true. We have demonstrated several means (specifically our new results on pH and temperature dependencies) the control the propelling speed, from remarkable acceleration to complete termination. Indeed, the *direction* of propelling is pre-determined here.

3) *What is really the new message here remains obscure.*

Reply: We have tried to make our message clearer by better emphasizing its unique aspects and adding new experiments and theory, which shed more light on intriguing aspects of the mechanism revealed here. Briefly, two highlights were put at the front of the revised version:

1. The macro-propelling of these motors is driven by a fundamentally new mechanism, i.e., the translation of a *molecular* chirality into particle's controlled macro-propelling. Macroscopic asymmetry of the particles is not needed.
2. Due to of pH-controlled Coulombic expelling of molecules, the motors have remarkable propelling performance, such as their velocities, cargo loads, etc.

Prof. Shachar Richter
Department of Materials Science and Engineering
Faculty of Engineering
&
University Center for Nanoscience and Nanotechnology
Tel Aviv University, Tel-Aviv, 69978, Israel

פרופ. שחר ריכטר
המחלקה למדע והנדסה של חומרים
הפקולטה להנדסה
&
המרכז לננומדע וננוטכנולוגיה
אוניברסיטת תל אביב, תל אביב 689978 ישראל

טל: (972)-36405711
פקס: (972)-36405612
srichter@tauex.tau.ac.il

Even the aspect of towing cargo has been explored much more quantitatively. Hence there is no real novel aspect in the paper (except for the quite obvious finding that the chirality of the motor depends on the chirality of the molecules used).

Reply: As already mentioned above, we showed here (for the first time) how molecular chirality (not particle chirality), can be dynamically converted into macroscopic unidirectional rotation of a symmetric particle. The correlation between molecular chirality and a particle's propelling direction may seem intuitively obvious. However, it turns out that the scientific problem is not as simple, bearing intriguing aspects that should draw future attention in related communities.

As for the cargo-related performances, besides their applicative importance, considerable motivation is raised here for achieving a better understanding of the observed capabilities and 'why these motors are so remarkable in cargo, speed, and fuel-capacity aspects'.

4) The paper is inconsistent. The model used is in the overdamped Stokes regime which is the regime of small Reynold numbers, see equation 1] in the SM (by the way Equation [2] is wrong and has probably a missing ")"). But on the other hand, the particles are macroscopic. Hence the theory needs a significant upgrade.

Reply: Following the suggestion of the referee, we have improved and upgraded the theoretical part, first by adding molecular dynamics simulations and, second, by improving the hydrodynamic model. Consistency is now accomplished in each of the two, while their complementarity allows a much more detailed understanding of the (rather complex) phenomenon under study.

Note specifically that the Reynolds numbers for the two examples analyzed with the hydrodynamic model are estimated to be 0.4 and 2 (as now explained in the SI). These values are typically considered to be within the Stokes (creeping flow) regime. The Reynolds number depends on the particles' size, mobility and the solute's dynamical viscosity. In our case, and following Figure 3, it is clear that this (Stokes) approximation reasonably reproduces the experimental trajectories and velocities. This improved analytical model is useful (see SI) in getting the correct order-of-magnitude estimates for the hydrodynamical loads (forces and torques) exerted on the particle. Obviously, it is incapable of accounting for the effect of molecular chirality on the actual particle trajectory. Therefore, to gain a further understanding

Prof. Shachar Richter
Department of Materials Science and Engineering
Faculty of Engineering
&
University Center for Nanoscience and Nanotechnology
Tel Aviv University, Tel-Aviv, 69978, Israel

פרופ. שחר ריכטר
המחלקה למדע והנדסה של חומרים
הפקולטה להנדסה
&
המרכז לננומדע וננוטכנולוגיה
אוניברסיטת תל אביב, תל אביב 689978 ישראל

טל: (972)-36405711
פקס: (972)-36405612
srichter@tauex.tau.ac.il

of the prevailing driving mechanism, we have added new atomistic modeling of the system, thus revealing intriguing aspects of the motor's dynamics.

5) There are statements that molecules are "ejected" and the picture is that "tangential momenta are exchanged". How can this happen in a liquid (water) without taking solvent flow effects into account?

Reply: According to the presented model, chiral molecules leave predominantly from certain crystal facets and spread over the water surface. In the related text, 'tangential momenta exchanges' refer to the condition of momentum conservation within the combined system of ejected molecule and its mother particle (not the solvent). Indeed, solvent flow effects are necessarily encountered in our experiments. However, they are proven to be secondary, as compared to the dominant molecule-based mechanism.

6) Maybe the self-propulsion mechanism has a significant phoretic part. Why can phoretic driving mechanisms be excluded?

Reply: The described mechanism could certainly be characterized as also having a phoretic component. In particular, the chemical part of phoretic mechanisms is now expressed and treated in our new atomistic model. Additional phoretic mechanisms, e.g., magnetic and electrical, are presently under study.

7) In Figure 3C and 3D 2d trajectories are shown but not analyzed. Are these real mathematical helices?

Reply: These helices are **numerically computed trajectories** (not mathematical) and indeed are predominantly 2D curves.

Prof. Shachar Richter
Department of Materials Science and Engineering
Faculty of Engineering
&
University Center for Nanoscience and Nanotechnology
Tel Aviv University, Tel-Aviv, 69978, Israel

פרופ. שחר ריכטר
המחלקה למדע והנדסה של חומרים
הפקולטה להנדסה
&
המרכז לננומדע וננוטכנולוגיה
אוניברסיטת תל אביב, תל אביב 689978 ישראל

טל: (972)-36405711
פקס: (972)-36405612
srichter@tauex.tau.ac.il

Reviewer #2 (Remarks to the Author):

The authors present a system where molecular chirality and relative chemical reactions result in a large-scale chiral active motion. The active particles are made of silver complex with chiral molecules, which placed at the air-water interface are gradually released producing a linear and angular momentum on the particle, which moves of chiral active motion. The same particles are used also to carry around some cargos.

The idea of the work is interesting especially when it draws a connection between nanoscale molecular chirality and large-scale chiral active motion. However, there are several points that are unclear and need to be addressed:

- 1. It's not clear how the chirality of the silver particle itself influence this overall process. It'd be necessary to study this aspect theoretically extending the theory as well as experimentally with increased statistics and with silver complexes with controllable shapes.*

Reply: We would like to thank the Reviewer for this idea. In the revised version of our work, we have expanded the theoretical section and added detailed *molecular dynamics* simulations to the hydrodynamic model, which we believe provides a new explanation of the experimental observations. Overall, the crystallite structure resembles its building-blocks chirality and, thus, affects the self-propulsion mechanism. We currently do not have the proper means to cast or synthesize particles with controllable shapes, but we could definitely try to do it in the future. In the revised version, we added a new set of experimental data and related statistics.

- 2. It's unclear how this work can lead to the concrete applications mentioned in the article. It'd be necessary to either demonstrate one such application, or to indicate a clear pathway towards such application.*

Reply: This is clearly a great idea; however not necessarily a trivial task to do. Some way on that path was already accomplished here, but we prefer to focus on gaining more understanding of the observed phenomena. In a follow-up work, we plan to examine a wider class of possible applications of these newly-found extraordinary motor towing characteristics.

Prof. Shachar Richter
Department of Materials Science and Engineering
Faculty of Engineering
&
University Center for Nanoscience and Nanotechnology
Tel Aviv University, Tel-Aviv, 69978, Israel

פרופ. שחר ריכטר
המחלקה למדע והנדסה של חומרים
הפקולטה להנדסה
&
המרכז לננומדע וננוטכנולוגיה
אוניברסיטת תל אביב, תל אביב 689978 ישראל

Tel: (972)-36405711 .טל.
Fax: (972)-36405612 .פקס.
srichter@tauex.tau.ac.il

These issues would need to be addressed very carefully with new theory and experiments before I can feel confident to recommend publication in Nature Communications.

Reply: In line with the referee's comment, our revised text was improved significantly by (1) expanding the hydrodynamical model (thus resulting in a favorably good agreement with measured particle trajectories), and (2) even more importantly, by including new molecular dynamic simulations that provide a much better physical insight into the prevailing driving mechanism. On top of those (3) there is a significant amount of additional experiments with their own interesting results, including the pH dependence which clearly correlates with our theoretical inputs.

Reviewer #3 (Remarks to the Author):

The paper by Carmeli et al. reports on self-propulsion of millimeter-sized objects at the water-air interface. The motion of such artificial motors, which are made of silver-organic complexes, results from the release of alkaloid molecules into the interface and completely stops when the concentration of alkaloid molecules saturates on the water surface. Depending on the type of the ejected molecules (cinchonine or cinchonidine) the motors can perform a persistent circular motion with a clockwise or anti-clockwise sense of rotation. Remarkably, the typical velocities achieved by the motors are of the order of several millimeters per second, whereas they are able to tow heavy cargo particles. Although the idea of designing efficient self-propelling motors have been receiving an increasing attention in various fields of both fundamental and applied research, the analysis and the results presented in this manuscript are scarce and do not provide a clear explanation of how the shape and size of the motors or the chirality of the released organic molecules can be employed to perform specific tasks at fluid-fluid interfaces. In addition, I have several concerns with regard to the data analysis, the modelling, and the interpretation of the results, which are listed below:

Reply: We have made considerable improvements in the experimental and theoretical parts. Please see below.

1) The paper lacks solid evidence of the physical origin of the rotational motion of the motors. The authors attribute the sense of rotation of the motors to the chirality of the ejected alkaloid molecules. However, a close inspection of table S1 of the Supplemental Information reveals that a significant fraction of the motors of each batch does not move as expected. In addition, the number of tested motors per batch (at most 15) is too small to correlate the rotational motion to the chirality of the molecules. The authors should at least improve the statistics of their experiments in order to draw reliable conclusions.

Prof. Shachar Richter
Department of Materials Science and Engineering
Faculty of Engineering
&
University Center for Nanoscience and Nanotechnology
Tel Aviv University, Tel-Aviv, 69978, Israel

פרופ. שחר ריכטר
המחלקה למדע והנדסה של חומרים
הפקולטה להנדסה
&
המרכז לננומדע וננוטכנולוגיה
אוניברסיטת תל אביב, תל אביב 689978 ישראל

טל: (972)-36405711
פקס: (972)-36405612
srichter@tauex.tau.ac.il

Reply: We gained much more statistical data with many additional experiments, now included in this revised and expanded version. Our new experiments validate the molecular chirality-based mechanism's significance (and dominance).

Note that we have also developed a new microscopic model based on molecular dynamics simulations, that provide a clear evidence that the released molecules can drive these motors asymmetrically. It also sheds some new light and gives a better idea how this motion could be controlled either by the types of the released molecules released or even the solvent pH. The new idea of being able to control the particle rotation by pH alone is highly interesting and relate these observations to several potential applications.

2) What is the role of temperature in the circular motion of the motors? For camphor flakes self-driven by Marangoni flows, it has been reported that the temperature strongly affects the frequency of revolution of the motion along the circular orbits [Nakata, S. et al., Langmuir 13, 4454–4458 (1997)]. The authors must specify under which temperature conditions they carried out their experiments.

Reply: We thank the reviewer for this comment. The issue was expanded and *new* temperature-dependent measurements are reported in the revised version. Not surprisingly, we found that at low temperatures, the rotation stops. We also measured the activation barrier for these rotations.

3) Did the authors investigate the effect of the motor shape on the resulting circular motion, e.g. Without this kind of analysis, it is impossible to disentangle the role of the Marangoni flow, the chirality of the motor due to its own shape and the chirality of the released molecules on the torques that give rise to persistent rotational movement.

Reply: So far, we have little control, if at all, on motors' shapes. In fact, as they are very irregular initially and, also, **undergo fast changes during the motion** (upon releasing bunches of molecules), the shape-related mechanisms tend to downgrade our statistics. The fact that our statistics does, after all, give a positive correlation - is **a strong proof** for the validity and dominance of the molecular level-based mechanism.

Prof. Shachar Richter
Department of Materials Science and Engineering
Faculty of Engineering
&
University Center for Nanoscience and Nanotechnology
Tel Aviv University, Tel-Aviv, 69978, Israel

פרופ. שחר ריכטר
המחלקה למדע והנדסה של חומרים
הפקולטה להנדסה
&
המרכז לננומדע וננוטכנולוגיה
אוניברסיטת תל אביב, תל אביב 689978 ישראל

טל: (972)-36405711
פקס: (972)-36405612
srichter@tauex.tau.ac.il

4) *The details about the model are not clear. In the SI, the authors mention briefly that they used a generalized Stokes approach based on the grand-resistance matrix to numerically find trajectories that resemble the experimental ones. However, they do not explain how to obtain the corresponding matrix elements, and how they are related to the alleged microscopic mechanism that produces the self-propulsion and self-rotation of the motors. In that sense, the model does not provide any useful information that elucidates how to exploit the proposed mechanism for particle steering, cargo towing, etc.*

Reply: We have now described in more detail (see SI) how we derived the matrix elements used to obtain Figure 3. As mentioned earlier, this model does not take into account the role of chirality, however with reasonable phenomenological inputs, it does reproduce within good accuracy the experimentally measured trajectories and velocities of the particles. The computed values are also compatible with the estimates of hydrodynamic forces of the order of a few nanonewtons, that are fixed in the frame of reference of the particles.

Importantly, in order to account for the driving mechanism that produces the self-propulsion and self-rotation of the motors, we have now added a whole new part of extensive molecular dynamics.

5) *The authors must thoroughly check the writing of the paper. They often refer to the SI for many details about the experiments and the model, but the needed information is missing there or is not clearly explained. Some references of the bibliography do not correspond to the main text (e.g., 21, 22 and 23). The SI contains a large amount of figures (S1, S2, S3, S5) that are never mentioned in the paper, and some of the figure and equation numbers are wrong (e.g. Figure X, Eq. A11, etc.)*

Reply: All typos and inconsistencies have been corrected

Sincerely,

Professor Shachar Richter

Prof. Shachar Richter
Department of Materials Science and Engineering
Faculty of Engineering
&
University Center for Nanoscience and Nanotechnology
Tel Aviv University, Tel-Aviv, 69978, Israel

פרופ. שחר ריכטר
המחלקה למדע והנדסה של חומרים
הפקולטה להנדסה
&
המרכז לננומדע וננוטכנולוגיה
אוניברסיטת תל אביב, תל אביב 689978 ישראל

טל. (972)-36405711
פקס. (972)-36405612
srichter@tauex.tau.ac.il

Reviewers' Comments:

Reviewer #2:

Remarks to the Author:

I think that the additional theoretical and experimental studies have strengthened the case for the publication of this study. I believe it now adds an intriguing phenomenon (coupling molecular chirality with macroscopic chiral active motion). In my opinion, these results are interesting enough to be published in Nat Commun.

Reviewer #3:

Remarks to the Author:

The authors have made a remarkable effort to address all the comments of my first report, and have substantially improved the quality of the manuscript. They have carefully explored and characterized additional effects on the unidirectional rotational motion of the chimots, such as pH and temperature, thus justifying the importance and novelty of the propelling mechanism. By performing molecular dynamics simulations and by extending the hydrodynamic model, they have also improved the theoretical description of the investigated system, which nicely support their experimental findings, especially those unveiling the physical origin of the unidirectional rotation of the chimots depending on the chirality of the released molecules. Therefore, I endorse publication of this paper

Reviewer #4:

Remarks to the Author:

General comments:

The novel version of the paper provides an added value concerning the modeling aspects in small Reynold numbers, and more importantly the inconsistency with the proposed experimental validation are now credible.

Replies to the authors comments:

The authors have improved and upgraded the theoretical part by adding molecular dynamics. The reviewer thinks that it is the right way to demonstrate the atomistic effects. In order to take into account the effect of molecular chirality on the actual particle trajectory, the authors provided molecular dynamics simulations using NAMD software. The Figure 3 provides details of the different conversion mechanisms in order to gain a deep understanding. The reviewer agrees that the proposed Stokes model approximation reasonably reproduces the experimental trajectories and velocities.

The reviewer has some minor comments related to the proposed simulation results:

- 1) The authors claim that the provided MD and the hydrodynamic model allows a much more detailed understanding of the phenomenon under study. However, it is not well explained how the simulations have been conducted. Are the both models connected together in a sequential way (multiscale and multiphysic approach?) or both models are separated? In the second case, the models are not connected and then cannot reproduce the experimental phenomena.
- 2) The reviewer is convinced that the described mechanism as also a phoretic component in relation to the chemical part of phoretic mechanisms. The atomistic model provides good estimates but it is not well demonstrated the influence of the phoretic part.
- 3) The influence of protonation is well described. There is a significant pH dependence which clearly correlates with the proposed theoretical inputs.

Dear Dr. Kreutzer

First, we would like to thank you for considering our manuscript for publication.

We would also like to thank the referees for their positive input.

Below please find our reply to the referees

Sincerely

Prof Shachar Richter, behalf of the authors

Reviewer #2 (Remarks to the Author):

I think that the additional theoretical and experimental studies have strengthened the case for the publication of this study. I believe it now adds an intriguing phenomenon (coupling molecular chirality with macroscopic chiral active motion). In my opinion, these results are interesting enough to be published in Nat Commun.

We thank the referee for the recommendation

Reviewer #3 (Remarks to the Author):

The authors have made a remarkable effort to address all the comments of my first report, and have substantially improved the quality of the manuscript. They have carefully explored and characterized additional effects on the unidirectional rotational motion of the chimots, such as pH and temperature, thus justifying the importance and novelty of the propelling mechanism. By performing molecular dynamics simulations and by extending the hydrodynamic model, they have also improved the theoretical description of the investigated system, which nicely support their experimental findings, especially those unveiling the physical origin of the unidirectional rotation of the chimots depending on the chirality of the released molecules. Therefore, I endorse publication of this paper

We thank the referee for the recommendation

Reviewer #4 (Remarks to the Author):

General comments:

The novel version of the paper provides an added value concerning the modeling aspects in small Reynold numbers, and more importantly the inconsistency with the proposed experimental validation are now credible.

Replies to the authors comments:

The authors have improved and upgraded the theoretical part by adding molecular dynamics. The reviewer thinks that it is the right way to demonstrate the atomistic effects. In order to take into account the effect of molecular chirality on the actual particle trajectory, the authors provided molecular dynamics simulations using NAMD software. The Figure 3 provides details of the different conversion mechanisms in order to gain a deep understanding. The reviewer agrees that the proposed Stokes model approximation reasonably reproduces the experimental trajectories and velocities.

The reviewer has some minor comments related to the proposed simulation results:

1) The authors claim that the provided MD and the hydrodynamic model allows a much more detailed understanding of the phenomenon under study. However, it is not well explained how the simulations have been conducted. Are the both models connected together in a sequential way (multiscale and multiphysic approach?) or both models are separated? In the second case, the models are not connected and then cannot reproduce the experimental phenomena.

We thank the referee for the comment- We added a relevant paragraph at the supplementary information (see the end of SI) discussing the connection between these two models with regard to evaluating the force and torque exerted on a chimot. In particular, we show that the results of the atomistic simulations can be roughly used to justify the forces used in the hydrodynamic model, thus providing a support for the driving mechanism of the chimots.

“... “...Using this scheme, it is possible to explicitly determine the values of F/β_{xx} , $T/\beta_{\varphi\varphi}$, $\beta_{x\varphi}/\beta_{xx}$, and $\beta_{xx}/\beta_{\varphi\varphi}$ resulting in the numerical simulations depicted in Fig. 3f (including a comparison with experiments), which include the trajectories obtained from numerically integrating the exact velocity expression. For the first chimot, we find $F/\beta_{xx} = 0.24$ nN, $T/\beta_{\varphi\varphi} = -0.039$ nN·mm, $\beta_{x\varphi}/\beta_{xx} = 0.11$, and $\beta_{xx}/\beta_{\varphi\varphi} = 0.24$. For the second chimot, we find $F/\beta_{xx} = 1.08$ nN, $T/\beta_{\varphi\varphi} = 0.39$ nN·mm, $\beta_{x\varphi}/\beta_{xx} = 0.41$, and $\beta_{xx}/\beta_{\varphi\varphi} = 0.077$. Realizing that some noise may still exist in the system, for example due to Brownian motion or the irregular solvation of the chimots, the experimental trajectories slightly differ from the computed ones, as seen especially in the case of the second chimot of Fig. 3f.

Although the above simplified hydrodynamic model doesn't capture the microscopic mobility mechanisms of the chimots, it does provide a useful insight into the kinematical behavior of the chimots. To begin with, it shows that the motion of chimots is compatible with standard propulsion mechanisms, where the acting force is always oriented in the same direction with respect to a body-fixed coordinate system (such as the gradient force due to surface tension under uneven solvation of the chimot). Assuming that the geometrical factors β_{xx} and $\beta_{\varphi\varphi}$ are of the same order of magnitude as those for a perfect sphere, we can deduce that the hydrodynamic forces and torques exerted on the chimots are of the order of a few nN and nN/mm, respectively. The proposed model also helps us to explain the fact that there is no clear correlation was observed in the experiments between the speed and size of the chimots. The force depends linearly on r ,

while the torque depends quadratically on r , therefore canceling any dependence of the speed on the size, if second-order terms are neglected.

We could try to use the MD simulation results to provide a rough estimate of the driving forces acting on a single chimot to justify the forces used in the above hydrodynamic model. The calculated energies presented in Fig. 3b show that the $(-1\ 0\ 0)$ and $(1\ 0\ 0)$ charged facets release molecules to the surrounding water, with a slight difference in their release rates (roughly 10%). For simplicity, consider that the surface tension of water drops to 50% of its original value of $\gamma_{\text{facet}} = 36\ \text{nN}/\mu\text{m}$ ($\gamma_{\text{water}} = 72\ \text{nN}/\mu\text{m}$) when fully saturated with amphiphilic cin molecules. We can also assume that the surface coverage above a particular facet is roughly proportional to its ejection rate of the cin molecules. This rate is proportional to the difference in binding energies to water and the crystal, ΔE_{facet} . Consequently, the surface tension above a particular facet can be estimated from $\gamma_{\text{facet}} = \gamma_{\text{water}} - c \cdot (\gamma_{\text{water}} - \gamma_{\text{saturated}}) \cdot \Delta E_{\text{facet}}$, where c accounts for the diffusion (removal) of released molecules. If the transport of released molecules is fast, but there is still a 10% difference in release rates between the facets, we can estimate the differences in surface tension across the crystal to $\Delta\gamma_{\text{crystal}} = 3.6\ \text{nN}/\mu\text{m}$. From $\Delta E_{\text{facet}} = \gamma_{\text{facet}} \cdot L \cdot \Delta x$, we can estimate the energy change of a liquid surface of γ_{facet} that is in contact with the facet of a length L and that moves in the orthogonal direction by Δx . The force acting on this facet, $F_{\text{facet}} = \Delta E_{\text{facet}} / \Delta x = L \cdot \gamma_{\text{facet}}$, produces a total force of $\Delta F_{\text{crystal}} = L \cdot \Delta\gamma_{\text{crystal}} = 9\ \text{nN}$, which is acting on a crystal with the side length of $L = 2.5\ \mu\text{m}$, that is exposed to the surface tension difference of $\Delta\gamma_{\text{crystal}} = 3.6\ \text{nN}/\mu\text{m}$. A chimot with a radius of $r=1\ \text{mm}$ can accommodate $N \sim 1,250$ of such crystals with adequate separations, giving a torque of $\tau = N \cdot \Delta F_{\text{crystal}} \cdot r = 11\ \mu\text{N}\cdot\text{mm}$. These values are about 3 orders of magnitude larger than our hydrodynamical estimates for the forces (4.5 nN and 18 nN) and torques (1 nN.mm and 10 nN.mm) acting on the two particles (crystals) of radius (0.32mm and 0.62mm) used in our Stokes simulations, resulting in the trajectories depicted in Fig. 3(d). The large difference between these forces can be related to the much smaller difference in the concentration of the molecules around the crystallites (constant c), a smaller asymmetry in crystallites positioned at the chimot surface (different orientations and submergences), and the complex shapes of chimots, further decreasing the number of active crystallites. At the same time, one can expect that when these parameters optimized one can significantly increase the driving of such systems". “

2) The reviewer is convinced that the described mechanism as also a phoretic component in relation to the chemical part of phoretic mechanisms. The atomistic model provides good estimates but it is not well demonstrated the influence of the phoretic part.

The MD simulations depicted in Fig.3 indicate that the released molecules tend to accumulate at higher concentration at one side of the microcrystal. This results in a diffusiophoretic driving due to the induced concentration gradient around the floating chimots (see sentence added to the main text)

“Note also that the MD simulations depicted in Fig. 3 indicate that the releases molecules tend to accumulate at higher concentration on one side of the microcrystal, resulting in a diffusiophoretic driving exerted on the floating chimots due to concentration gradients.”

3) The influence of protonation is well described. There is a significant pH dependence which clearly correlates with the proposed theoretical inputs.

We thank the referee supporting this point